

# Investigating the relationships among lung function variables in chronic obstructive pulmonary disease in men

Ming-Lung Chuang[1,2] and I-Feng Lin[3]

[1] Division of Pulmonary Medicine, Department of Internal Medicine, Chung Shan Medical University Hospital, Taichung, Taiwan
[2] School of Medicine, Chung Shan Medical University, Taichung, Taiwan
[3] Institute of Public Health, National Yang Ming University, Taipei, Taiwan

## ABSTRACT

**Background**. In patients with chronic obstructive pulmonary disease (COPD), the independent contributions of individual lung function variables to outcomes may be lower when they are modelled together if they are collinear. In addition, lung volume measurements may not be necessary after spirometry data have been obtained. However, these hypotheses depend on whether forced vital capacity (FVC) can predict total lung capacity (TLC). Moreover, the definitions of hyperinflation and air trapping according to lung function variables overlap and need be clarified. Therefore, the aim of this study was to evaluate the relationships among various lung function parameters to elucidate these issues.

**Methods**. Demographic data and 26 parameters of full lung function were measured in 94 men with COPD and analyzed using factor and correlation analyses.

**Results**. Factor analysis revealed five latent factors. Inspiratory capacity (IC)/TLC and residual volume (RV)/TLC were most strongly correlated with all other lung volumes. IC/TLC, RV/TLC, and functional residual capacity (FRC)/TLC were collinear and were potential markers of air trapping, whereas TLC%, FRC%, and RV% were collinear and were potential markers of hyperinflation. RV/TLC $>0.4$ (or IC/TLC $<0.4$) was comparable with the ratio of forced expiratory volume in one second ($FEV_1$) and FVC $<0.7$. FVC% and $FEV_1$% were poorly correlated with TLC%. The correlation study showed that TLC%, RV/TLC, and $FEV_1$% could be used to represent individual latent factors for hyperinflation, air trapping, inspiration, expiration, and obstruction. Combined with diffusion capacity%, these four factors could be used to represent comprehensive lung function.

**Conclusions**. This study identified collinear relationships among individual lung function variables and thus selecting variables with close relationships for correlation studies should be performed with caution. This study also differentiated variables for air trapping and lung hyperinflation. Lung volume measurements are still required even when spirometry data are available. Four out of 26 lung function variables from individual latent factors could be used to concisely represent lung function.

Corresponding author
Ming-Lung Chuang,
yuan1007@ms36.hinet.net

## INTRODUCTION

Lung function variables including residual volume (RV)/total lung capacity (TLC), inspiratory capacity (IC)/TLC, RV% predicted and forced expiratory volume in one second percentage predicted (FEV$_1$%) have been reported to be well correlated with the outcomes of patients with chronic obstructive pulmonary disease (COPD) including physiological deterioration, dyspnea, exercise intolerance, acute exacerbations, and mortality (*O'Donnell, Revill & Webb, 2001*; *Nishimura et al., 2002*; *Casanova et al., 2005*; *Tantucci et al., 2008*; *Vassaux et al., 2008*; *Zhang et al., 2013*; *Chuang, Huang & Su, 2015*; *Shin et al., 2015*). Correlations between lung function variables and outcomes have been reported sporadically, such as associations between IC/TLC and RV/TLC and prognosis, (*Nishimura et al., 2002*; *Casanova et al., 2005*; *Shin et al., 2015*) exercise capacity, (*Vassaux et al., 2008*; *Zhang et al., 2013*; *Chuang, Huang & Su, 2015*) exertional dyspnea, (*Casanova et al., 2005*; *Zhang et al., 2013*; *Chuang, Huang & Su, 2015*) and dynamic lung expansion (*Zhang et al., 2013*; *Chuang, Huang & Su, 2015*) in patients with COPD. Moreover, IC/TLC and RV/TLC have been reported to have very similar Harrell's C-statistics in prognostic analysis (*Shin et al., 2015*). However, as the independent contributions of IC/TLC and RV/TLC to outcomes would be lower when they are modelled together if they are highly correlated, it is important to investigate the relationships among lung function variables.

Hyperinflation and air trapping are defined using lung volumes in percentages (%) combined with ratios of functional residual capacity (FRC), IC or RV and TLC (*Ruppel, 1991*; *Gagnon et al., 2014*; *Cohen, 2017*; *Vaz Fragoso et al., 2017*). However, these definitions are unclear and arbitrary and understanding the relationships among these variables may improve the definitions.

It has also been reported that TLC measurements are not necessary after spirometry data have been obtained. However, this will not be the case if forced vital capacity (FVC) cannot predict TLC, as obstructive ventilatory impairment may be combined with restriction in patients with COPD (*Dykstra et al., 1999*; *Nishimura et al., 2002*; *Calverley & Koulouris, 2005*; *Deesomchok et al., 2010*; *Labbé et al., 2010*; *Shin et al., 2015*). To resolve these three issues, this study aimed to thoroughly evaluate the relationships among lung function parameters using factor and correlation analyses.

## METHODS

### Study design

In this observational cross-sectional study, we analyzed lung function data from participants with COPD at Chung Shan Medical University Hospital. A total of 26 lung volume and capacity parameters were expressed in liters, %predicted, and ratio of TLC, FVC and SVC (Appendix 1). The Chung Shan Medical University Hospital Institutional Review Board (CS11144 and CS19014) approved this study, which was conducted in compliance with the Declaration of Helsinki.

## Participants

The diagnosis of COPD was made by board-certified pulmonologists according to the GOLD criteria (*GOLD Committees, 2017*). Participants with other thoracic diseases such as pleural diseases or thoracic cage disorders were excluded from this study (*GOLD Committees, 2017*). As few female participants meet the COPD criteria in Taiwan (e.g., 4% according to one study *Huang et al., 2018*), they were excluded from this study. Male adult participants with a post-bronchodilator $FEV_1/FVC$ of <0.7 (*GOLD Committees, 2017*) or 0.7–0.8 with a definite obstructive pattern in spirometry were enrolled (*Johns, Walters & Walters, 2014*). The latter definition was used for two reasons. First, extensive small airway disease can exist before it is detectable with a $FEV_1/FVC$ <0.7, and thus the concavity of expiratory flow-volume curve was considered (*Johns, Walters & Walters, 2014*). Second, some participants with increased small airway compressibility had a preserved $FEV_1/FVC$ but a reduced $FEV_1$/slow VC ratio (*Johns, Walters & Walters, 2014*; *Saint-Pierre et al., 2019* (in press)). The exclusion criteria included a $FEV_1/FVC$ of 0.7–0.8 with an equivocal obstructive pattern or with a significant post-bronchodilator effect, i.e., increase in $FEV_1$ of >12% and 200 mL from baseline (*GINA Committees, 2017*) or bronchial asthma (*GINA Committees, 2017*) diagnosed by the board-certified pulmonologists. Bronchial asthma was excluded because the proportion of lung subdivisions is different in these two diseases (*Dykstra et al., 1999*). Informed consent was obtained from each participant by them signing the consent form.

## Measurements
### *Pulmonary function testing*

Cigarette smoking, drinking coffee, tea, or alcohol, and taking medications were not permitted 24 h before any test. Bronchodilators were not administered within 3 h for short-acting beta agonists and 12 h for long-acting beta agonists before the tests. $FEV_1$, TLC, and RV were measured using spirometry and body plethysmography (MasterScreen$^{TM}$ Body; Carefusion, Wuerzburg, Germany) in accordance with the currently recommended standards (*Miller et al., 2005a*; *Miller et al., 2005b*; *Wanger et al., 2005*). The best of three technically satisfactory readings was used (*ATS/ERS S, 2002*; *Miller et al., 2005a*; *Miller et al., 2005b*). All of the spirometry data were obtained before and after inhaling 400 µg of fenoterol HCl. Post-dose measurements were performed 15 min after inhalation. Static lung volume data and diffusing capacity for carbon monoxide ($D_LCO$) data measured using the single-breath technique were obtained before inhaling fenoterol. For details of lung subdivision measurements, please refer to Appendix 2. Simple volume calibration was conducted using a 3-L syringe before each test. Accuracy checks for body plethysmograph mouth flow and pressure and box pressure were conducted daily. We have previously reported the predicted values currently used at our institute. The predicted values are in line with our previous report (*Chuang, Lin & Wasserman, 2001*). The reason that we did not use lower limit of normal as the criterion of airflow limitation was that we did not have reference equations using post-bronchodilator $FEV_1$ and FVC for the Taiwanese population.

## Statistical analysis

All of the data were checked for normal distribution by the Kolmogorov–Smirnov test. Data were summarized as mean ± standard deviation or median (25th–75th percentiles) when appropriate. We used factor analysis to evaluate the correlated pulmonary function variables and identify the latent factors. We extracted the initial set of factors using the principal-component method based on the Kaiser criterion, then 1-more and 1-fewer factor models were also evaluated. The factors were then rotated using an orthogonal transformation method (VARIMAX in SAS) to assess the interpretability of the factors. The numbers of factors determined in the final model was based on biological plausibility. Pearson's or Spearman's correlation coefficients were further used when appropriate for quantifying the pair-wise relationships among the pulmonary function variables. All statistical analyses were performed using SAS statistical software (SAS Institute Inc., Cary, NC, USA). Statistical significance was set at two-sided $p < 0.05$.

## RESULTS

A total of 94 male participants (mean age 68.1 ± 7.2 years) with COPD were enrolled after excluding 10 participants (Table 1 and Fig. 1). Five of these 10 participants who were excluded were diagnosed with bronchial asthma including three females and another five had spirometry data that did not meet the inclusion criteria. Only five of the 94 participants had a $FEV_1/FVC$ ratio 0.7–0.8 with a definite obstructive pattern. Most of the participants had moderate airflow obstruction (Table 1).

Up to 26 lung function parameters were presented in absolute values, %predicted, and as the ratio of lung volume or capacity and TLC or FVC for each patient. In normal participants, lung volume in liters was closely related to body height, sex, and age and hence it was usually presented with %predicted. Therefore, the following variables were omitted from correlation analysis: TLC, FRC, IC, IRV, ERV, RV, FVC, SVC, and $FEV_1$ in liters and $D_LCO$ in mL/min/mmHg.

### Factor analysis

A preliminary model with four factors were selected by the Kaiser criterion and then the 3-factor and 5-factor models were also evaluated. The five latent factor model was identified according to biological plausibility and this model explained 92% of total variation (Table 2). The communalities were generally high (12 of 16 were >0.9, three were between 0.75 and 0.9, and only one was 0.4). Factor 1 was highly related to inspiration and moderately to air trapping; factor 2 was highly related to lung volumes and moderately to air trapping; factor 3 was highly related to expiration and moderately to air trapping; factor 4 was highly related to airflow obstruction; and factor 5 was highly related to diffusion capacity.

### Pearson's or Spearman's correlation coefficients for lung volume subdivisions

All $r^2$, the coefficient of determination for Pearson's correlations, indicated the proportion of variance in one variable explained by variation in the other. Individual variables

**Table 1  Demographics and lung function in 94 participants with chronic obstructive pulmonary disease.**

|  | Mean | SD |
|---|---|---|
| Age, years | 68.1 | 7.2 |
| Height, cm | 164.5 | 5.8 |
| Weight, kg | 61.3 | 9.6 |
| Body mass index, kg/m$^2$ | 22.6 | 3.2 |
| Cigarette smoke[a], pack . year | 45 | 35.5–60[e] |
| TLC% predicted[a], % | 116 | 103–137[e] |
| FRC% predicted[a], % | 143 | 35 |
| FRC/TLC[a] | 0.71 | 0.08 |
| RV% predicted[a], % | 170 | 55 |
| RV/TLC[a] | 0.56 | 0.09 |
| Expiratory reserve volume, ERV% predicted[a], % | 82 | 55–121[e] |
| ERV/TLC[a] | 0.15 | 0.08 |
| IC% predicted[a], % | 87 | 26 |
| IC/TLC[a] | 0.29 | 0.08 |
| Inspiratory reserve volume, IRV% predicted[a], % | 78 | 47 |
| IRV/TLC[a] | 0.15 | 0.09 |
| $D_L$CO% predicted, %, unadjusted for Hemoglobin[b] | 76 | 26 |
| FVC% predicted, % | 83 | 22 |
| FEV$_1$% predicted, % | 54 | 19 |
| Mild, FEV$_1$% ≥80%[d], n, % | 8 (9) | |
| Moderate, 50% ≤ FEV$_1$%<80%[d], n, % | 47 (50) | |
| Severe, 30% ≤ FEV$_1$%<50%[d], n, % | 30 (32) | |
| Very severe, FEV$_1$%<30%[d], n, % | 9 (10) | |
| FEV$_1$/FVC | 0.51 | 0.12 |
| SVC% predicted[c], % | 82 | 70–107[e] |

**Notes.**

TLC, total lung capacity; L, liter; FRC, functional residual capacity; RV, residual volume; IC, inspiratory capacity; $D_L$CO, diffusing capacity for carbon monoxide; FVC, forced vital capacity; FEV$_1$, forced expired volume in one second; SVC, slow vital capacity.

[a] $n = 92$.
[b] $n = 88$.
[c] $n = 91$.
[d] *GOLD Committees (2017)*.
[e] median (25th–75th percentiles).

involving %predicted were strongly correlated with each corresponding variable in ratio of TLC (Table 3, all $r^2 = 0.38 - 0.85$, all $p < 0.0001$). TLC%, FRC%, and RV% were highly correlated ($r^2 = 0.59 - 0.74$). RV% was the best single parameter correlated with $D_L$CO%, although only with a negative moderate correlation ($r^2 = 0.22$, $p < 0.0001$). FRC/TLC and IC/TLC were reciprocal ($r^2 = 0.98$) and highly correlated with RV/TLC ($r^2 = 0.37$-$0.38$), and were most frequently correlated with all of the other lung subdivisions in %predicted and ratio of TLC. IC% and IRV% were approximately collinear ($r^2 = 0.90$).

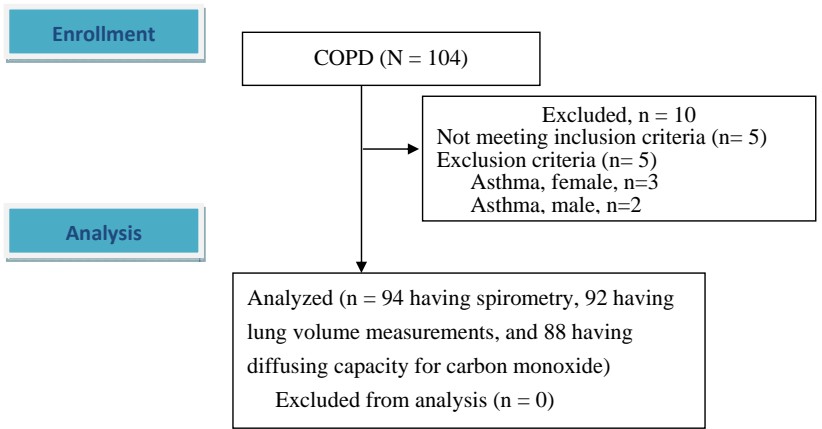

**Figure 1** **Flow diagram.** A total of 104 participants with chronic obstructive pulmonary disease were screened. 94 patients were enrolled and analyzed after excluding 10 participants. Of the 94 participants, two participants missed the lung volume measurements, and six participants missed diffusing capacity for carbon monoxide. The differences in demographic data and spirometry between the six and 88 participants were not statistically significant except body weight and body mass index ($54.5 \pm 3.0$ kg versus $61.7 \pm 9.7$ kg, $p < 0.001$ and $20.9 \pm 1.3$ kg/m$^2$ versus $22.7 \pm 3.2$ kg/m$^2$, $p < 0.05$, respectively).

**Table 2** **Factor analysis showing rotated component matrix[a].**

| | Factors | | | | |
| | 1 | 2 | 3 | 4 | 5 |
|---|---|---|---|---|---|
| TLC%p | 0.27 | **0.93** | 0.17 | −0.10 | −0.10 |
| FRC%p | −0.20 | **0.94** | 0.11 | −0.13 | −0.08 |
| FRC/TLC | *−0.68* | 0.65 | 0.10 | −0.21 | −0.07 |
| RV%p | −0.07 | **0.89** | −0.32 | −0.13 | −0.16 |
| RV/TLC | −0.39 | *0.53* | *−0.67* | −0.16 | −0.14 |
| IC%p | **0.96** | .011 | 0.08 | 0.09 | −0.03 |
| IC/TLC | *0.67* | *−0.66* | −0.10 | 0.21 | 0.04 |
| IRV%p | **0.94** | 0.07 | 0.10 | 0.01 | −0.03 |
| IRV/TLC | **0.86** | −0.19 | 0.03 | −0.01 | 0.13 |
| ERV%p | −0.18 | 0.22 | **0.91** | −0.04 | 0.14 |
| ERV/TLC | −0.24 | 0.02 | **0.91** | −0.03 | 0.09 |
| D$_L$CO%p | 0.07 | −0.25 | 0.19 | 0.18 | **0.92** |
| FVC%p | 0.48 | −0.09 | *0.76* | 0.12 | −0.04 |
| FEV$_1$%p | 0.37 | −0.26 | *0.54* | *0.65* | 0.06 |
| FEV$_1$/FVC | −0.00 | −0.22 | −0.04 | **0.94** | 0.15 |
| SVC%p | *0.57* | 0.06 | *0.77* | 0.08 | 0.04 |

**Notes.**

For all abbreviations, please refer to Table 1. Extraction method: Principal component analysis. Rotation method: Varimax with Kaiser normalization.

[a]Rotation converged in 5 iterations.

Bolded number indicating the important variables (arbitrarily defined as value $>0.85$) in that factor, Italic number indicating the variables with moderate importance (arbitrarily defined as value $> 0.50$) in that factor.

**Table 3  Pearson Correlations (r) Pair-Wise Deletion between lung volumes in 92[a] participants with obstructive airway disease.**

| | TLC%# | FRC% | FRC/TLC | RV% | RV/TLC | ERV%# | ERV/TLC | IC% | IC/TLC | IRV% | IRV/TLC | $D_L$CO% |
|---|---|---|---|---|---|---|---|---|---|---|---|---|
| TLC%# | | 0.86§ | 0.44§ | 0.77§ | 0.26** | −0.27** | NS | 0.33** | −0.46§ | 0.41§ | NS | −0.28** |
| FRC% | | | 0.75§ | 0.87§ | 0.52§ | 0.32** | NS | NS | −0.76§ | NS | −0.28** | −0.34** |
| FRC/TLC | | | | 0.62§ | 0.62§ | 0.29** | 0.29** | −0.63§ | −0.99§ | −0.53§ | −0.65§ | −0.29** |
| RV% | | | | | 0.71§ | NS | −0.23* | NS | −0.63§ | NS | −0.22* | −0.47§ |
| RV/TLC | | | | | | −0.45§ | −0.57§ | −0.44§ | −0.61§ | −0.38⑤ | −0.51§ | −0.43§ |
| ERV%# | | | | | | | 0.92§ | NS | −0.30** | NS | NS | NS |
| ERV/TLC | | | | | | | | NS | −0.27** | NS | NS | 0.21* |
| IC% | | | | | | | | | 0.62§ | 0.95§ | 0.77§ | NS |
| IC/TLC | | | | | | | | | | 0.52§ | 0.65§ | 0.27* |
| IRV% | | | | | | | | | | | 0.77§ | NS |
| IRV/TLC | | | | | | | | | | | | 0.21* |

**Notes.**

TLC, total lung capacity; FRC, functional residual capacity; RV, residual volume; ERV, expiratory reserve volume; IC, inspiratory capacity; IRV, inspiratory reserve volume; $D_L$CO, diffusing capacity for carbon monoxide; %, % predicted.

#Spearman correlation coefficient.

§$p < 0.0001$.

†$p < 0.001$.

**$p < 0.01$.

*$p < 0.05$.

[a]Two of 94 participants missed lung volume measurements.

NS, non-significant.

**Table 4** Pearson Correlations (r) Pair-Wise Deletion between lung volumes and spirometry in participants with obstructive airway disease.

| | FVC% | FEV$_1$% | FEV$_1$/FVC | SVC%%[#] |
|---|---|---|---|---|
| FVC% | | 0.77[§] | 0.04 | 0.86[§] |
| FEV$_1$% | | | 0.63[§] | 0.62[§] |
| FEV$_1$/FVC | | | | −0.03 |
| TLC%[#] | 0.20[*] | −0.11 | −0.32[**] | 0.39[†] |
| FRC% | −0.09 | −0.35[†] | −0.36[†] | 0.09 |
| FRC/TLC | −0.33[**] | −0.50[§] | −0.36[†] | −0.28[**] |
| RV% | −0.29[**] | −0.48[§] | −0.35[†] | −0.20 |
| RV/TLC | −0.65[§] | −0.67[§] | −0.28[**] | −0.67[§] |
| ERV%[#] | 0.43[§] | 0.27[**] | −0.07 | 0.58[§] |
| ERV/TLC | 0.45[§] | 0.29[**] | −0.03 | 0.50[§] |
| IC% | 0.51[§] | 0.44[§] | 0.08 | 0.62[§] |
| IC/TLC | 0.32[**] | 0.49[§] | 0.36[†] | 0.25[*] |
| IRV% | 0.50[§] | 0.37[†] | 0.01 | 0.63[§] |
| IRV/TLC | 0.32[**] | 0.29[**] | 0.06 | 0.51[§] |
| D$_L$CO% | 0.19 | 0.34[**] | 0.29[**] | 0.18 |

**Notes.**

TLC, total lung capacity; FRC, functional residual capacity; RV, residual volume; ERV, expiratory reserve volume; IC, inspiratory capacity; IRV, inspiratory reserve volume; D$_L$CO, diffusing capacity for carbon monoxide; FVC, forced vital capacity; FEV$_1$, forced expired volume in one second; SVC, slow vital capacity; %, % predicted.

[#] Spearman correlation coefficient.
[§] $p < 0.0001$.
[†] $p < 0.001$.
[**] $p < 0.01$.
[*] $p < 0.05$.
$n = 94$ having spirometry, 92 having lung volume measurements, and 88 having D$_L$CO.

## Pearson's or Spearman's correlation coefficients for spirometry variables

FVC% and SVC% were highly correlated with FEV$_1$% ($r^2 = 0.38 - 0.74$, Table 4); however, they were not correlated with FEV$_1$/FVC ($r^2 = 0.00$-$0.00$). Interestingly, FEV$_1$% and FEV$_1$/FVC were related ($r^2 = 0.40$).

## Pearson's or Spearman's correlation coefficients between spirometry and lung volume subdivisions

FVC% and FEV$_1$% were modestly or not correlated with TLC% (Table 4, $r^2 = 0.01$–$0.04$) despite being correlated with most of the subdivisions in %predicted and ratio of TLC. Of 89 participants with FEV$_1$/FVC <0.7, 81 (91%) had IC/TLC <0.4 and 85 (96%) had RV/TLC ≥0.4.

Based on these results, TLC%, RV/TLC (or FEV$_1$/FVC), FEV$_1$%, and D$_L$CO% were selected from the five factors as representative of lung function.

# DISCUSSION

## Factor analysis and Pearson's or Spearman's correlation coefficients

Factor analysis is a good method to identify unobservable factors from a large number of observed variables, thereby allowing variables to be used to estimate a lower number

of latent factors. In this study, we identified five latent factors, i.e., those highly related to inspiration, expiration, lung volumes, airflow obstruction, and diffusion, respectively (Table 2). However, lung volumes and inspiration and expiration volumes were also mixed with air trapping, suggesting that hyperinflation and air trapping overlapped (see below). In further analysis, Pearson's or Spearman's correlation coefficients revealed the lung function variables associated with hyperinflation and air trapping (Table 3).

However, it is possible that the small sample size would not meet the usual heuristics for principle component analysis or factor analysis, and that this would mean it was difficult to explain the implications of the non-normality and outliers in the data. Although the sample size of the present study was small at 88, the influence of the small sample size depends on the data characteristics. Previous studies have shown that even a small sample size of fewer than 50 participants can yield results with small distortions under data conditions of high loading, low number of factors, and high number of variables (*De Winter, Dodou & Wieringa, 2009*). When communalities are high, the sample size tends to have less influence on the quality of factor solutions than when communalities are low *Hogarty et al., 2005*. We only had limited data; however, the communalities were generally high (12 of 16 were >0.90, three were between 0.75 and 0.90, and only one was 0.40) in our final factor model. Spearman's correlation coefficients were applied for the pairs involving non-normality (Tables 3 and 4) and the results were not substantially different from those using the Pearson's correlations. Therefore, we assumed that the potential non-normality might not substantially influence the current results of factor analysis.

In addition, one may argue the rationale for using the Kaiser criterion for selecting a preliminary model in this study. While both the Kaiser criterion and parallel analysis use information from eigenvalues of the correlation matrix, the latter can be used to determine more consistently the number of factors, especially in small samples. As appropriate factor retention would depend on biological plausibility rather than purely on statistical consideration, we used the Kaiser criterion to determine the preliminary model, and then selected a five-factor final model (as stated in the previous sections). One of the purposes of this study was to explore different domains (latent factors) related to individual lung function parameters. Rotation makes latent factors more interpretable. We used VARIMAX, an orthogonal rotation that did not change the communalities and the total variance explained (still 92%) while preserving correlations between variables (*Rencher & Christensen, 2012*).

## Lung volume subdivisions

It has been frequently reported that both IC/TLC and RV/TLC are related to prognosis, exercise capacity, exertional dyspnea, and dynamic lung expansion in patients with COPD. Similarly, IC/TLC and RV/TLC have been reported to have very similar Harrell's C-statistics in prognostic analysis (0.81 and 0.80, respectively, Table 4 of reference (*Shin et al., 2015*). We hypothesize that IC/TLC and RV/TLC have similar Harrell's C-statistics because of their collinearity.

RV% was the largest and the most frequent responder following bronchodilation in a previous report (*Deesomchok et al., 2010*). In the present study, it showed the best inverse but moderate correlation of all lung function variables with $D_LCO\%$.

Definitions of pulmonary hyperinflation and air trapping of the lung are inconsistent in the literature, for example: static hyperinflation = hyperinflation at rest (*Gagnon et al., 2014*) = IC/TLC <0.25 (*Casanova et al., 2005*) = RV/TLC (*Nishimura et al., 2002*; *Budweiser et al., 2014*) $\geq$0.4 (*Albuquerque et al., 2006*; *Shin et al., 2015*) or >0.35 plus an increased TLC (*Ruppel, 1991*) or >0.3 plus RV% $\geq$120% (*Labbé et al., 2010*); air trapping = RV/TLC >0.35 plus a normal TLC (*Ruppel, 1991*) or RV% $\geq$120% (*Deesomchok et al., 2010*). To some extent, these definitions overlap RV, IC and TLC and their derivatives. In addition, the relationship between RV/TLC and TLC% is inconsistent in previous reports (*Deesomchok et al., 2010*; *Vaz Fragoso et al., 2017*). Despite knowing that factor three involved lung volumes, we further evaluated the relationships among all lung subdivisions with Pearson's or Spearman's correlation, and found that RV%, FRC%, and TLC% were highly correlated ($r^2 = 0.59$–$0.74$), and that RV/TLC, FRC/TLC and IC/TLC were also highly correlated ($r^2 = 0.37$–$0.98$). In contrast, the correlations of RV/TLC, FRC/TLC and IC/TLC with TLC% were only $r^2 = 0.08$, $0.18$, and $0.20$, respectively.

RV, FRC, and TLC in %predicted are all related to static lung volume (*Gagnon et al., 2014*; *Vaz Fragoso et al., 2017*). RV% is mainly composed of FRC%, (*Dykstra et al., 1999*) and FRC% is mainly composed of TLC% in COPD. This is consistent with a previous report in which RV% was highly correlated with FRC% ($r = 0.9$) (*Deesomchok et al., 2010*). Therefore, we recommend that RV%, FRC%, and TLC% can be used as biomarkers for hyperinflation, and that RV/TLC and IC/TLC (reciprocal FRC/TLC) can be used as biomarkers for air trapping, even though these two types of biomarkers are closely related. It makes sense that the IC/TLC triad is more sensitive than RV% (*Albuquerque et al., 2006*; *Zhang et al., 2013*) and $FEV_1\%$ in relation to exercise capacity, as the IC/TLC triad, RV% and $FEV_1\%$ belong to different factors, denoting that air trapping plays a more important role in exercise capacity than hyperinflation and airflow obstruction (*Zhang et al., 2013*).

## Spirometry and lung volume subdivisions

The need for simultaneous measurements of lung volume and spirometry is controversial. $FEV_1/FVC$, a biomarker of airway obstruction, was significantly correlated with the biomarkers for air trapping and hyperinflation in this study (Table 4). Airway obstruction can easily be assessed using spirometry. Therefore, additional measurements of static lung volume add little to the clinical interpretation. This is consistent with the study by Dykstra et al., in which 87% of 1,872 patients with reduced vital capacity had a high RV/TLC, and only 10% had a low TLC% (*Dykstra et al., 1999*). This concept was further confirmed in their report as $FEV_1\%$ was reported to predict RV%, RV/TLC, and TLC% ($r = -0.76$, $-0.66$ and $-0.33$, respectively, all $p < 0.0001$) (*Dykstra et al., 1999*). Another study reported that a low FVC% with a low $FEV_1/FVC$ ratio could be used as a marker of obstructive ventilation with "pseudo-restriction" (*Aaron, Dales & Cardinal, 1999*). However, 10% of the patients in Dykstra's study had a low TLC%, suggesting the coexistence of a mixed type (obstructive-restrictive impairment) (*Dykstra et al., 1999*). In addition, another study reported that 8%

of asthmatics had restricted ventilation (*Miller & Palecki, 2007*). Gardner et al. reported that restriction of the lung may interfere with classifying the severity of obstruction in patients with mixed obstructive-restrictive lung disease according to $FEV_1\%$ adjusted for TLC%. This adjustment resulted in the downgrading of 83% of their patients to a lesser degree of obstruction (*Gardner, Ruppel & Kaminsky, 2011*; *Aaron, Dales & Cardinal, 1999*) reported that a reduced FVC could only predict a reduced TLC by 40–50%. Moreover, regarding the effect of bronchodilators on COPD, measurements of lung volume response may be superior to those of flow response (*Deesomchok et al., 2010*; *McCartney et al., 2016*). Hence, we disagree with the notion that lung volume measurements are not necessary for patients with reduced VC, (*Dykstra et al., 1999*) as FVC% and $FEV_1\%$ were modestly or not correlated with TLC% in the present ($r^2 = 0.04$ and $0.01$, Table 4), and previous studies (*Dykstra et al., 1999*; *McCartney et al., 2016*) and were not correlated with expandable lung volumes (Table 4, IC%, IRV%, ERV%, and IRV/TLC and ERV/TLC). The smaller the $FEV_1\%$, the larger the RV% and RV/TLC, even though TLC% had probably yet to change (*Vaz Fragoso et al., 2017*).

$FEV_1\%$ was also linearly related to IC (*Deesomchok et al., 2010*) and VC% (*Nishimura et al., 2002*; *Deesomchok et al., 2010*) in this study, as $FEV_1$ was $0.76 \pm 0.26$ of IC and IC was $0.71 \pm 0.20$ of FVC. FVC%, SVC%, and $FEV_1\%$ were highly correlated in this study but not with $FEV_1/FVC$, which may be because a reduction in FVC may result in a normal or mildly reduced $FEV_1/FVC$ due to pseudo-restriction (*Saint-Pierre et al., 2019* (in press)).

In summary, TLC% cannot be replaced by FVC% and RV/TLC is most frequently and most strongly correlated with other lung function variables and consistent with $FEV_1/FVC$. $FEV_1\%$ is a marker of the severity of COPD. $D_LCO\%$ alone represented factor 5. Although these four variables could be used to represent all 26 full lung function variables, we do not conclude that the additional variables should not be collected/examined/included in models.

## Study limitations

Diagnostic instability of COPD diagnosis has been reported in approximately 20% and 10% of individuals with mild and moderate airflow obstruction, respectively, after 4–5 years of follow-up (*Aaron et al., 2017*). However, most (92%) of our participants had moderate to very severe airflow obstruction; therefore, we estimated that 6.7% of our participants probably had diagnostic instability. Selecting an appropriate reference value is important, e.g., TLC or FVC has been reported to be 12% lower in African Americans than in Caucasians (*Lapp et al., 1974*). In this study, we arbitrarily reduced the reference values in the literature by 10–15%, as surveillance reports on reference values were unavailable and to consistently follow our previous reports (*Chuang, Lin & Wasserman, 2001*). Lung size has been reported to be different between Chinese participants from southern China and northern China; however, it is difficult to trace the study participants in Taiwan, as some ancestors came from both northern and southern China. Nevertheless, the use of correlation analysis may have reduced this potential bias. In Guangzhou, China, the estimated prevalence rates of GOLD stage 2 or higher COPD in females and males are approximately 5% and 10%, respectively (*Mannino & Buist, 2007*). Although we did not
have prevalence data of COPD in Taiwan, 96% of the cohort with COPD were male and only 4% were female (*Huang et al., 2018*). The data from China are quite different from ours, which may be related to the biomass fuel smoke exposure in the Chinese female population (*Mannino & Buist, 2007*). Although our study population was small compared to previous reports (*Aaron, Dales & Cardinal, 1999*; *Dykstra et al., 1999*), our patients had COPD alone, whereas previous studies have enrolled COPD patients with various other lung diseases. This may raise concerns that the relationships between lung volumes and their ratios to TLC among the study participants of previous studies may be different from those in our study (*Ruppel, 2012*). In addition, our data were all obtained before bronchodilator inhalation, as pharmacological interventions may alter the relationship between volume and capacity. Hence, the findings of this study should be interpreted with caution when extrapolating to patients after bronchodilator inhalation. Furthermore, only two participants did use ultra-long-acting beta agonists in this study, so that the duration of bronchodilator withdrawal for most of the participants might be appropriate. The relationships among lung function variables in the current study may also have been different if the lung volume and capacity were expressed in a way other than % predicted. For example, a previous study reported lung volume and capacity were standardized with cubed height or adjusted with multiple linear regression analysis with adjustments for least square mean and adjustments of spirometry with $z$-scores (*Vaz Fragoso et al., 2017*). However, these adjustments have not yet been widely used in clinical practice, and further studies are needed to clarify this issue. In the present study, %predicted and ratio of lung volume or capacity and TLC or FVC were used instead of using absolute value in liters or mL/min/mmHg. This may have reduced confounding caused by body height, sex, and age. In addition, anemia is a factor that should be considered for adjustment when predicting $D_L CO$ (*Macintyre et al., 2005*). However, this may be a minor issue, as the hemoglobin level was $14.4 \pm 1.7 g/dL$ and 83.2% of our participants had normal hemoglobin levels ($\geq 13$ g/dL) and the adjusted coefficient for predicted $D_L CO$ was $0.99 \pm 0.05$ (5th percentile–95th percentile: 0.89–1.06). Lastly, COPD has many phenotypes, however we did not specify these phenotypes in this study.

## CONCLUSIONS

This study identified several collinear relationships among individual lung function variables. Selecting multiple variables with close relationships for correlation studies should be performed with caution. This study also differentiated variables for air trapping and lung hyperinflation. Lung volume measurements are necessary even when spirometry data are available. We identified four of 26 lung function variables from individual latent factors that could be used to concisely represent lung function.

## APPENDIX 1

The 26 lung function variables with their units.

| | |
|---|---|
| 1 | TLC, L |
| 2 | TLC% predicted, % |
| 3 | FRC, L |
| 4 | FRC% predicted, % |
| 5 | FRC/TLC |
| 6 | RV, L |
| 7 | RV% predicted, % |
| 8 | RV/TLC |
| 9 | Expiratory reserve volume, L |
| 10 | Expiratory reserve volume, ERV% predicted, % |
| 11 | ERV/TLC |
| 12 | IC, L |
| 13 | IC% predicted, % |
| 14 | IC/TLC |
| 15 | Inspiratory reserve volume, L |
| 16 | Inspiratory reserve volume, IRV% predicted, % |
| 17 | IRV/TLC |
| 18 | Diffusion capacity for cabon monoxide, $D_LCO$, mL/min/mmHg |
| 19 | $D_LCO$% predicted, % |
| 20 | Forced vital capacity, FVC, L |
| 21 | FVC% predicted, % |
| 22 | Forced expiratory volume in one second, $FEV_1$, L |
| 23 | $FEV_1$% predicted, % |
| 24 | $FEV_1$/FVC |
| 25 | Slow vital capacity, SVC, L |
| 26 | SVC% predicted, % |

## APPENDIX 2

TLC was defined as IC plus FRC (*Wanger et al., 2005*). Thoracic gas volume was measured by panting at a rate of 1 Hz when the flow was occluded at the end of stable expiration. FRC was calculated from thoracic gas volume after adjusting for stable tidal volume. IC was measured from the end of stable expiration to the top of TLC as far as possible after panting. $V_T$ was averaged from several stable breaths at rest, and IRV was measured from IC minus the averaged $V_T$. ERV was measured as SVC minus IC, and RV was measured as FRC minus ERV. Predicted IC was calculated as predicted TLC minus predicted FRC, and predicted ERV was calculated as predicted FRC minus predicted RV (*Deesomchok et al., 2010*).

### Funding

Chung Shan Medical University Hospital provided financial support (CSH-2012-C-23, CSH-2019-C-30). There was no additional external funding received for this study. The funders had no role in study design, data collection and analysis, decision to publish, or preparation of the manuscript.

### Grant Disclosures

The following grant information was disclosed by the authors:
Chung Shan Medical University Hospital: CSH-2012-C-23, CSH-2019-C-30.

### Competing Interests

The authors declare there are no competing interests.

### Author Contributions

- Ming-Lung Chuang conceived and designed the experiments, performed the experiments, analyzed the data, contributed reagents/materials/analysis tools, prepared figures and/or tables, authored or reviewed drafts of the paper, approved the final draft.
- I-Feng Lin analyzed the data, contributed reagents/materials/analysis tools, authored or reviewed drafts of the paper, approved the final draft.

### Human Ethics

The following information was supplied relating to ethical approvals (i.e., approving body and any reference numbers):

The Chung Shan Medical University Hospital Institutional Review Board (CS11144 and CS19014) approved this study.

### Data Availability

The raw measurements are available as a Supplemental File.

### Supplemental Information

Supplemental information for this article can be found online at http://dx.doi.org/10.7717/peerj.7829#supplemental-information.

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
