# Peer review of "Investigating the relationships among lung function variables in chronic obstructive pulmonary disease in men"

_PeerJ, doi:10.7717/peerj.7829_

## Round 0.1 · original submission · Major Revisions

The reviewers have identified some challenges for you to consider and I have added my comments below. I think it might be difficult to overcome all of these, especially given the sample size and non-normal distributions of the measures in your data set, but if you feel you can fully address the reviewers’ and my comments, I would be pleased to see a revised version of the manuscript. All of the reviewers’ comments and mine below would need to be addressed with point-by-point responses and changes to the manuscript as appropriate.

The manuscript has the appealing goal of reducing the number of lung function variables needed, but this is difficult to justify clinically from a purely data reduction perspective. Variables cannot be ignored only because of their correlation with other variables or linear combinations of other variables. It is still possible for such variables to be important prognostically, diagnostically, or otherwise, although higher correlations do indeed make their independent contributions to such models likely to be lesser on average. Using varimax rotation here means that highly correlated variables can still form orthogonal factors in any case. The manuscript would be much stronger with an appropriate endpoint or endpoints so that this use of these variables could be examined with a clinical goal in mind. As it is, the focus on data reduction is acceptable, but this limits the conclusions that can be drawn. You could note that a set of variables explains some suitable high percentage of the total variation across a larger set of variables, but you could not then conclude that the additional variables should not be collected/examined/included in models based solely on those empirical results. Making this point would require a different study design, one that looked at clinical outcomes and then used, for example, likelihood ratio tests or an information theoretic approach to argue whether these was further value in adding some variables given a set of variables already being in the model.

Note that factor analysis (FA) and principle components analysis (PCA) are very different approaches, despite sharing mathematical foundations. The use of FA implies a set of latent variables, which seems at odds with your goal of data reduction, but which is precisely the goal of PCA. Can you explain why you have chosen FA here?

The Kaiser criterion (Line 135) is known to be suspect and parallel analysis is the usual standard when exploring the number of factors or components. Why have you use this approach? Also, why have you rotated the factors? And why varimax? This allows variables that are highly correlated to appear independent after rotation, which seems to go against your thesis here.

Note that Pearson’s correlations assume linear associations. You would need to justify this assumption in the manuscript and include appropriate diagnostics if this approach continued to be used. Given the presence of outliers in your data, this makes these correlations, and the FA, potentially highly sensitive to a small number of values. While looking at pairwise correlations can help in preparing for PCA or FA, and sometimes in understanding components or factors, it cannot resolve any “dilemmas” found using multivariate analyses (Lines 177–178). The issue of cross-loadings (of which there are several in Table 2) needs very careful attention.

A data dictionary would be helpful in understanding your data set and in obtaining the same data set as was used for analysis. Like Reviewer #1, I am confused about the actual sample size used. For the variables listed in Table 1, in your data set of n=97 rows, after removing n=3 with sex=2 (presumably female?), there appear to be only n=94 (not n=95) patients with data. Of these, each variable has between 88 (DLCO) and 94 observations with values, so the analysis would appear to be based on only n=88 patients (this is without any further deletion of the n=2 with FEV1/FVC values over 0.75, being 0.77 and 0.80, which appear to be automatically excluded based on Line 111). Figure 1 appears to be based on a CONSORT diagram but this is inappropriate for this study design (see Reviewer #1’s comments also). There are no allocations to groups, nor is there loss to follow-up as they were not followed over time (this point is in fact noted in the Figure 1 legend). More useful would be a flowchart that lists the actual reasons for exclusion (see Reviewer #1’s comments again).

Some of the variables, perhaps especially notably FRC/TLC but most of the variables display some univariate non-normality and/or outliers, are not normally distributed and this has implications for summary statistics based around means and SDs, and also for looking at the structure of the data using techniques that assume multivariate normality for some results and that are sensitive to outliers. How have these issues been addressed?

It’s not explicitly explained in the manuscript how you go from n=26 lung function variables (Line 147) to the n=16 variables analysed. By only modelling n=16 variables, as far as I can see, this wouldn’t in itself permit the statement that only 4 of the 16 can stand in for all 26 even setting aside the issues raised above (Lines 252–254). Note that in Table 3, FEV1, FVC, and SVC are listed in the legend but not included in the table.

This sample size (apparently n=88 or n=86) would not meet the usual heuristics for PCA or FA with even 16 variables and this analysis on such a small data set would need to be well justified (with appropriate references) along with explaining the implications of the non-normality and outliers in the data (these characteristics would normally lead to much larger sample size requirements). Note also that these associations could be confounded by age, BMI, smoking, etc., and this would need to be discussed, and ideally addressed. You allude to related points to this in Lines 271–275 of the discussion.

The criteria used to identify the “important variables in that component” and the “variables with moderate importance in that component” should be made clear in Table 2.

There is work to be done on the writing style and you may wish to contact a copyediting service (for example, the service offered by PeerJ) to address this. I’ve made a few copyediting comments below about the abstract, along with some other points about the results and conclusions there, but I have not provided detailed feedback about the writing in the body of the manuscript (although Reviewers #1 and #2 have kindly noted some aspects that they feel need editing). This will need to be addressed in a resubmitted version.

For the abstract, I think the background could be shortened considerably (126 words on the background is a lot).

I would recommend softening Line 39, perhaps to “To our knowledge, they have not been considered together.” as absolute statements about a lack of research on a topic are difficult to verify completely.

Lines 39–40: It’s not entirely clear to me what you mean by “To avoid the duplicate reports on these correlation studies…” There is a similar awkward phrase “may be duplicate and meaningless” on Lines 57–58.

Line 42: Do you mean “relationships” (with an “s”) here? Also Line 57.

Lines 43–44: This doesn’t logically follow as written.

Line 53: Take care with absolute statements. Not being correlated is not the same thing as there being no evidence for a correlation.

Lines 54–55: The clause “and were essential for full lung function report.” is interpretation and really should be covered in the abstract’s conclusions (although, I note that a similar point is already there and so this clause can simply be deleted from the results).

Lines 59–60: I think you need to clarify what the necessity is for here. Similarly, in the final sentence, what does “essential” mean here? If these variables did not provide additional prognostic or diagnostic (or some other) information on top of other variables, they would not be essential just because they are not strongly correlated with other variables and, similarly, strongly correlated variables can still be invaluable.

Reviewer 1 ·

Basic reporting

There is only one figure in the manuscript.
The figure is of medium quality and contains unnecessary fragments - e.g. “give reasons”, when analyzed group is zero.
It raises my reservations. Figure 1 is described as a flow diagram of 107 people with COPD, while the graphics and text show that the number of registered people was 105. How many people actually were?
It should be more precise and clear.

Experimental design

The inclusion and exclusion criteria you refer to at work should be clearly and precisely described. Now I cannot see exclusion criteria at all.

In line 111 you write about FEV1/FVC ratio, but I cannot be sure if it is pre or postbronchodiator measurement.
In line 111 you write you have patients enrolled with FEV1/FVC ratio between 0.7-0.75.
We should know how many. Can you check if these patients have FEV1/FVC ratio below lower limit of normal (LLN)? How you comment it if you are asked for GOLD document point of view?

Aaron et al (Am J Respir Crit Care Med. 2017 Aug 1;196(3):306-314. doi: 10.1164/rccm.201612-2531OC.) described the problem of diagnostic instability and reversals of Chronic Obstructive Pulmonary Disease diagnosis in individuals with mild to moderate airflow obstruction. Why haven’t you taken it into consideration?

In line 111 you write about definite obstructive pattern in spirometry. You should define it adding correct reference.
Importantly, I would suggest that the information in Table 1 should contain more detailed information characterizing the group of patients. I advise that the table show what were the number of groups for each interval of degrees of obstruction that you use.

Mannino et Biust (Mannino DM, Buist AS. Global burden of COPD: risk factors, prevalence, and future trends. Lancet 2007;370:765–773) have shown that prevalence of COPD among men and women e.g. in China. Your enrolled group described in line 113 is deeply not consistent with above. Could you explain it, please?
If you decide analyze men COPD population only, you should think about changing the title by adding "in men".

In line 119 you write long-acting beta agonists were stopped 12 h before measurement. Haven’t your patients used ultra-LABA e.g. indacaterol?
In line 133 word “to” is used twice.
In line 144 you decide to exclude the patients because of co-existing asthma. It is deeply not precise. What kind of criteria have you used to do it. It should be described.
In line 145 and 146 you write about moderate airflow obstruction in most patients. You should precisely inform about numbers and you should define which division of obstruction you use giving adequate reference.

There is a lack of information if DLCO measurement was correlated with hemoglobin level.

Validity of the findings

no comment

Additional comments

Dear Author,

It was nice to get the opportunity to review the manuscript
"Investigating the relationships among lung function variables in chronic obstructive pulmonary disease".
I was reading it with interest however I need to share my honest opinion.

Undoubtedly, the weakness of this work is the size of the group. However, I am still open to analyze such small population. But in such case all aspects of the methodology of work must be perfect. This is not here in my opinion, what I have included in detailed comments.

For example, I have serious doubts that all patients actually had COPD. I do not know how asthma was diagnosed.
I see a lot of inaccuracies that require unquestionably improvement.

In my opinion, the manuscript has potential and should be published after the major amendments.

Sincerely,

Reviewer

·

Basic reporting

- English language (expression & grammar):
The following lines need revision: 41-42, 88-89, 210-211, 252 & 265-266.
- Literature references:
All citations within the text should be revised. Please note that when using “et al.”, only the name of the first author should be mentioned.
- Professional article structure:
Acceptable.

Experimental design

- The research question:
Acceptable.
- Investigation & ethical standard:
Acceptable.
- Methods:
Well described.

Validity of the findings

- The findings are original & valid. The conclusion is well stated.

Additional comments

- The third objective (lines 43-44) is not clear.
- The statements mentioned at lines 212-215: (… in relation to exercise capacity as RV/TLC, RV% and FEV1% belong to different components, denoting that air trapping played a more important role in exercise capacity than hyperinflation and airflow obstruction is not supported by convincing findings). Even the reference cited (Zhang Y et al., 2013) reports a different indicator (IC/TLC rather than RV/TLC).

---

## Round 0.2 · Minor Revisions

Thank you for your revisions and constructive responses. The one relying reviewer had no further comments. However, some of my comments on the previous version were not fully addressed.

I’m still not entirely convinced by your arguments around factor analysis versus principal components analysis, use of rotation, using Kaiser’s criterion (even with also looking at one more or one fewer factor; note that a reference and the use of fewer and more factors should be added around Line 126), or the ratio of variables to observations, but these are perhaps sufficiently subjective points of departure for an exploratory study, although I also strongly agree with Reviewer #1’s earlier comment that “…in such case all aspects of the methodology of work must be perfect.” and so I will encourage you to continue to think carefully about these matters, although I will not require further changes in response to them beyond the note about factor selection above.

However, the exact data and analyses using this remains difficult to follow (I’ll reiterate my previous request for a data dictionary). On Lines 88–90, you say: “As few female subjects meet the COPD criteria in Taiwan (i.e., 4% according to one study (Huang et al., 2018)), they were excluded from this study.” and the title has been revised to include “…in Men”. Without a data dictionary, I am again assuming that sex==2 represents those 3 women (with sex==1 for men, giving n=97 with this value). This, however, leaves n=88 men with DLCO values, not n=89 as you state, and n=94 (97-3) participants overall, not n=95 as you state. If these three “women” are not dropped, making the above quoted text appear to be misleading, this then gives n=89 for DLCO, but n=97 in total not n=95. I remain unable to reconcile your study numbers with the text in the manuscript and I would appreciate you doing three things: 1) adding a data dictionary (with all variables included) and checking your uploaded data to ensure that it reflects the data used in the most recent version of the manuscript, 2) explaining what numbers of participants are included for each outcome and for each analysis based on the final data set, and 3) modifying the manuscript as necessary to reconcile the data and text. I apologise if I am misunderstanding something about the data and/or analyses and if this is the cause of my inability to reconcile the data and results, but if this is the case, I am confident that other readers will also encounter the same challenge.

Note also that I cannot replicate Table 1 based on the data provided (which might be simply because of the different ns suggesting different data sets). I suspect that this might also be due in part because of values for “smoke” (these are normally numeric values, but include “Y” and “0.5ppd?” for sex==1, which is difficult to interpret, and a value of “?” for sex==2 that I cannot interpret but may not be important), but some other descriptive statistics are also not replicated with or without the sex==2 participants included. The data provided needs to allow the reader to replicate each and every one of the presented results and so these issues need to be resolved and the text made clear. In order to also make the data used clearer, please add a column showing the number of observations for each variable to Table 1. Please also add the number of observations used to each other table (For Tables 3 and 4, I am assuming that this will have to be a minimum–maximum number of observations with pairwise deletion used?)

I suggest checking that you can replicate every one of the presented results in the manuscript using the data file you have uploaded or will upload along with your revised manuscript. Providing SAS code would be an excellent way to ensure that everything is clear and replicable for me and any other readers.

I also suggest cleaning the data file to remove the blank column for “name”.

While I appreciate you adding a caveat about the non-normality of some variables, it would be much better to address this through more appropriate summary statistics (geometric means and geometric standard deviations for log-normally distributed variables, and medians and IQRs for other continuous variables would be reasonable choices). Similarly, rather than using Pearson’s correlations (Tables 3 and 4), using Spearman’s or one of Kendall’s correlations would be more appropriate in the presence of skew and/or outliers. Any scatter plot showing a non-statistically significant association between two variables can with the addition of a single data point (although this will need to be quite unusual) have a Pearson’s correlation coefficient in either direction with p<0.001, so it is not just marginal results that need to be interpreted with care. The pattern of results for Table 3 appears to differ, albeit slightly, between choices of correlation coefficient used (I haven’t checked Table 4) and so you need to use a method that can be justified for the data you have. Your factor analysis is still vulnerable to these issues in the data also, unless I am not identifying that some non-standard method was used, and factor structures can be arbitrarily changed through modifying small numbers of observations. I appreciate that the outlying values seen here (for example, for IRV % prediction) are not that extreme but they are extreme enough to require careful treatment so that findings can be generalised. There is an extensive literature on PCA and FA with non-normal data or outliers, including https://doi.org/10.1177%2F0146621618798669, https://doi.org/10.1007/BF02294711, https://doi.org/10.1007/BF02294108, https://doi.org/10.1111%2Fj.1467-9531.2008.00198.x, https://doi.org/10.1080/13504850802046989, https://doi.org/10.1080/02664763.2015.1005063,
https://doi.org/10.1016/j.talanta.2006.10.011, and https://doi.org/10.1016/j.csda.2007.05.024 but you will find much more than this written on this matter.

The explanation of which variables were excluded (Lines 140–144) doesn’t seem to me to make it entirely clear that all absolute values were excluded in favour of percentages of predicted. Could you also quantify the correlation(s) on Lines 141–142 that you are using to justify this removal? A quick check of the data set I have suggests r=0.49 for n=92 with TLC and height, which is only one quarter of the variation in TLC explained by variation in height, for example.

Related to this, I wouldn’t call r=0.79 “extremely collinear” (Line 152), this still leaves around 38% of the variation in each unexplained by the other variable. It might be useful in this section of the results (Lines 150–163) to consider the coefficients of determination (r-squared values) when judging how collinear/correlated variables are.

Reviewer 1 ·

Basic reporting

no comment

Experimental design

no comment

Validity of the findings

no comment

Additional comments

You have significantly improved the quality of work. The message is currently more readable and comprehensible.

---

## Round 0.3 · Minor Revisions

Thank you for your revised version of the manuscript. I think we are almost there for me being able to accept your manuscript.

I appreciate your explanation about FA versus PCA, however my point there was not the estimation approach, but rather the different models implied by the two approaches. This is an issue that I encounter as a biostatistician in several other fields also, particularly nutrition, where the literature often includes both approaches being used, ostensibly for the same purpose. Simply for completeness, what I mean is that PCA is a data summarization approach, aiming at reducing a set of variables into a smaller number of component scores that capture a sufficient amount of the variation in the original data; whereas FA implies a latent construct which is reflected in the measured variables (the direction being modelled here is latent factor scores -> observed variables, which is the opposite of PCA where the observed variables -> component scores). The use of PCA seemed to me to be a more natural model for what you are interested in (data reduction), rather than postulating latent variables describing lung function (which cannot be directly measured), although both approaches could certainly be argued for. Continuing with my comments on the statistical methods, Kaiser’s criterion is well known to misestimate the true number of factors (by extracting too many factors) and while I appreciate, and strongly agree with, the argument you make around theoretical considerations, looking at one more or one fewer factor following Kaiser’s criterion would not recover the true number of factors in a consistent manner. As I said last time, these are simply points for you to consider and if you are happy with your approach, then there is no need to make changes in response to these particular comments.

Regarding my other points, thank you very much for the updated data file and the addition of a data dictionary. This includes 94 men as described in the manuscript. I don’t see how, though, you can have n=89 (Line 185) for the factor analysis as DLCO is only available for n=88, so I assume this is a typo? Some of the other values in the manuscript also did not match my checking of the data, my apologies if I have misunderstood anything here, and I have noted these points below. You mention hemoglobin (Lines 318–319), but this did not seem to be in the data set. Could it be added?

I appreciate your other thorough responses to my questions. In some cases, some of that information should, I think, be included in the manuscript—in particular the process for selecting the number of factors.

Since the content seems to be very close to being finalized, I’ve checked over the writing carefully throughout the manuscript. There is some work needed on the language and I’ve made some specific comments below to correct some errors and suggested some ways to improve the flow of the manuscript. There are too many of these to leave for the proofing stage, I feel, so I’ll ask you to address these now. It is of course possible that I have missed some such points and you might wish to use a professional proofreading service as well or instead of my language-related comments. Please do feel free to decline any of my stylistic suggestions.

Specific comments:

Line 27: Assuming I am understanding your argument here, perhaps “…lower when they are modelled together if…” (adding “when they are modelled together”) as in univariable models, this could go either way. The same point applies to Lines 64–65. Otherwise, the reader could spend time wondering why this would be the case for univariable regression models or bivariate analyses.

In the abstract, I think readers will wonder how you got from “five latent components” (Line 35) to the four variables listed on Line 41 and the conclusion that “4 … lung function variables…could be used to concisely represent lung function.” Perhaps some additional information could be added here?

Line 35: I’d change “components” to “factors” on this line as this is the usual terminology for FA and the current wording might make readers wonder about PCA. See also Lines 42, 47, 127 (twice), 148 (twice), 149, 150, 151, 152, 170, 175, 176, 206, 228, 241, 278, 327, Table 2 caption and notes, and anywhere else I’ve missed this (but not when talking about the estimation approach). You could also use “latent variables” in some places if you wanted.

Lines 52–53: The % notation seems awkward here, e.g. the “(%)” on Line 52, which does not seem necessary, and the trailing “%” on Line 53 might work better as “…in one second percentage predicted (FEV1%)…”.

Line 63: I’d suggest “prognostic” rather than “prognosis” here. Also Line 214.

Line 64: See comment for Line 27 above.

Line 67: “…in % combined…” doesn’t work if “%” is read as “percentage”, which is more usual than “percentages”, which is often written “%s”. Perhaps “…in percentages (%) combined…”

Line 80: Rather than “subjects”, please use “participants”. This term is generally considered more respectful and indicates their contribution to the research rather than passive involvement. Also Lines 86, 88, 90, 91, 96, 136, 137 (twice), 139, 140, 143, 167, 187, 291, 292, 309, 319, 344, 345 (twice), 346 (twice), 348, Table 1 caption, Table 3 caption and notes, Table 4 caption, Flow diagram 1 notes, and anywhere else I might have missed this.

Line 90: By “i.e.” (“that is”) here, do you instead mean “e.g.”, i.e. “for example”?

Line 92: Missing space after “0.7” here.

Line 97: No space after period in “.The”

Lines 102–103: Do you mean: “Informed consent was obtained from each participant by them signing the consent form.” or “Informed consent forms were signed by each participant.”?

Line 116: You could delete “the” from “the Appendix 2.”

Line 119: You could delete “currently used at our institutes” from here to avoid the immediate repetition of this phrase on the previous line (although institute becomes institutes, so make sure the correct one is used).

Line 122: “for the Taiwanese population.” (adding “the”)

Line 125: “ranges” should be “range”, but since you don’t present IQRs later on, this should be “(25th–75th percentiles)”. See https://en.wikipedia.org/wiki/Interquartile_range for an explanation.

Lines 127–130: Your responses to my questions about the number of factors suggests a more involved process was used, which you only mention here in the discussion. Can you please explain the method here in sufficient detail for a reader to be able to hope to replicate it. This should also help the reader to better understand the results as they read that section before they get to the discussion.

Line 131: When you say “appropriate by quantifying” do you mean “appropriate for quantifying”?

Line 133: Assuming it is the case, I’d add “two-sided” just before “p < 0.05”.

Lines 137–138: Were women ever eligible? There were 3 women (sex=2) in the previous data set, not 2. The same point arises with Figure 1.

Line 148: Can you please use some of your responses to my queries about the determination of the number of factors to expand on this point, explaining what models were examined and how the final model was determined.

Line 155: My suggestion around r and R-squared was more intended to inform the interpretation of the magnitude of correlations (my comment was: “Related to this, I wouldn’t call r=0.79 “extremely collinear” (Line 152), this still leaves around 38% of the variation in each unexplained by the other variable. It might be useful in this section of the results (Lines 150–163) to consider the coefficients of determination (r-squared values) when judging how collinear/correlated variables are.”), but I’m also happy with your approach of reporting R-squared values throughout. I think that a short comment here, the first instance of this, might be useful for some readers though, e.g. “…all r2 [the coefficient of determination for Pearson’s correlation coefficients, indicating the proportion of variance in one variable explained by variation in the other]=0.38…” or a footnote to explain the concept, potentially in a little more detail.

Line 160: I’d insert “approximately” before “collinear” here as “collinear” has a precise mathematical definition that is an absolute one.

Line 163: “FEV” not “FVE” here (twice).

Line 167: Are you sure it’s 90, I get n=89 with FEV/FVC<0.7?

Line 167: Are you sure it’s 85, I get n=81 with both FEV/FVC<0.7 and IC/TLC<0.4 (91%). I suggest using integer percentages as with < 100 participants, each counts for over 1% and so the decimal place is not meaningful.

Line 167: Are you sure it’s 88, I get n=86 with both FEV/FVC<0.7 and RC/TLC>0.4 (97%).

Line 170: Do you mean “representative” rather than “representatives”?

Line 174: The point of latent variables is more that they are “unobservable” rather than simply “unobserved”. This would also make the contrast between the unobservable latent factors and the “observed variables” clearer to the reader.

Line 175: I’m not sold on the idea of “sorted” here. Table 2 shows evidence of some substantial cross loading. Perhaps you can find another way of expressing the point you’re making here, such as “…allowing variables to be used to estimate a lower number of latent variables.”

Line 181: “trapping” not “tapping”.

Line 182: I’m not sure about the “it is possible” here. The sample size definitely “would not meet some conventional heuristics for…” You might like to cite a few references for sample size heuristics here, although I do appreciate you work through some of the literature below.

Line 183: Do you mean “would make it difficult” (deleting “was”) or “would mean it was difficult” (changing “make” to “mean”) here?

Like 185: As far as I can tell, with DLCO only available for n=88, you cannot have used n=89 for the FA.

Lines 190–192: It would be nice if these were (also) included in the results section.

Line 194: “those using the” (“using” rather than “by”) would seem more usual wording.

Line 198: Perhaps “determine more consistently the number” (adding “ly” and changing “a” to “the”).

Line 200: Suggesting changing word order to “than purely on statistical”.

Line 201: “1-fewer” (factors are countable).

Line 203: I don’t think you need “additional” here, it implies (to me) that there was more than one 3-factor and more than one 5-factor model considered.

Line 204: Perhaps “satisfied” rather than “met” here.

Lines 204–205: It’s not clear, and this will presumably be explained back in the methods, why starting with a 4-factor model, using the plus/minus one approach to look at 3- and 5-factor models, would then lead to looking at a 6-factor model, unless the process was iterative, but then why would more than 6 factors ever be looked at in this case if the 6-factor model was not chosen, at least temporarily?

Line 209: You can delete “could be” here.

Line 214: Please keep the decimal places consistent, even if this requires adding a trailing zero or zeros. Also Line 232, and it might be worth checking the rest of the manuscript for any other inconsistencies.

Line 259: You have a double “%” here.

Line 272–273: Please provide the reader with information about what each number represents. For the (presumed) first SD, I get 0.2551, which would round to 0.26 and not 0.25. Similarly, for the second SD, I get 0.1953, which would round to 0.20 and not 0.19. Can you please check these values?

Line 276: Delete “is” in “TLC% is cannot”

Line 294: I’d perhaps add “potential” so that this reads “reduced this potential bias.” unless you are certain that the bias would exist otherwise.

Line 299: Rather than “smoke in the Chinese female population”, perhaps “smoke exposure in the Chinese female population”?

Line 304: I’m not sure what you mean by “our data in relationship study”.

Lines 311–312: I’m not sure what you mean by “adjustments for least square mean” as LS means is an estimation approach for linear regression and similar. Did you want to list variables that could be adjusted for here (age, sex, height)? Similar for adjustments with z-scores.

Line 316: There seems to be an extra space (or spaces) here.

Line 316: Do you mean gender (the social construct) or sex (biological state) here?

Line 320: Perhaps “(5th percentile–95th percentile: …)” here.

Line 323: Do you mean “clarified” or “identified”?

Lines 323–324: I’m not sure that “thus” can be justified (the first point does not directly lead to the second unless you do mean “identified” above). Otherwise, perhaps delete “and thus” and start a new sentence after where this was.

Line 330: I suggest using “Appendix 1” here and “Appendix 2” on Line 333 below to help the reader find the second appendix more easily.

Line 331: I’d append “with their units” or similar here.

Line 348: Rather than “insignificant”, I think you mean “not statistically significant”. Please give actual p-values here rather than inequalities and indicate the direction of the associations (knowing that weight and BMI varied between the groups isn’t useful without knowing which was heavier).

Table 1: The note should indicate that you are reporting “(25%-75% percentiles)” rather than “(interquartile ranges 25%-75%).” See comment for Line 125.

Table 2: The note should refer to “Table 1” not “Tables 1”.

Figure 1: The text “(give reasons)” is instructions not something to retain in the final version of the figure.

---

## Round 0.4 · accepted · Accept

Thank you for your revisions. I am delighted to accept your manuscript, although there are a small number of possible typos and inconsistencies that will need to be addressed at the proofing stage. These are listed below. Well done!

There may have been differences in our line numbers as there were two instances of “prognosis” that I suggested changing to “prognostic”, but you have made four such changes. The changes to Line 61 (“and RV/TLC and prognostic”) and Line 218 (“related to prognostic”) should have remained as “prognosis” (these are nouns), whereas the changes to Lines 65 and 220 which now use “prognostic” (an adjective describing the meaning of “analysis” in each case) are the two changes I had intended.

Line 132: “based on THE Kaiser criterion” (adding “the” before “Kaiser” as “criterion” is singular and not plural). The same applies to Line 153 (add “the” before “Kaiser”).

Line 149: You updated “gender” to “sex” on Line 324 but not here.

Lines 168–170: While it could remain here, I think that this explanatory sentence would be more useful earlier in the paragraph, perhaps starting on Line 161.

Line 168: There is a missing second decimal place in “were approximately collinear (r2=0.9).” which should be shown as a zero if this is the case.

Line 173: Same point for “and FEV1/FVC were related (r2=0.4).”

Line 215: The comma between “change the communalities” and “and the total variance explained” does not seem necessary.

Line 446: There seems to be a missing field here (“Methods of multivariate analysis. , A John Wiley & Sons,”) as indicated by the spurious comma, and the publisher would normally be given simply as “John Wiley & Sons Press” (but do check your copy of this book). Note that book titles are normally italicised in the same way that the journal titles are.

Line 448: Italicise book title here also.

Table 2: Could you make the decimal places consistent here also? (For example, replacing “-0.2” with “-0.20”).

Table 3: Same point as above, with “-0.3” seeming to be the only exception to two decimal places.

Table 4: Same point again with 0.5 appearing three times (once as a negative) and -0.2 once.